# Impact of the “Warning Signs Campaign” on Characteristics of Patients Presenting with Acute Coronary Syndrome (ACS) to Hospitals

**DOI:** 10.3390/ijerph191710700

**Published:** 2022-08-27

**Authors:** Eleanor Redwood, Karice Hyun, John French, Derek Chew, Leonard Kritharides, David Brieger

**Affiliations:** 1Department of Cardiology, Northern Beaches Hospital, Sydney, NSW 2086, Australia; 2School of Sciences, Faculty of Medicine and Health, University of Sydney, Sydney, NSW 2006, Australia; 3ANZAC Research Institute, Concord Repatriation General Hospital, Sydney, NSW 2137, Australia; 4Liverpool Hospital, Liverpool, NSW 2170, Australia; 5Department of Cardiology, Flinders University, Adelaide, SA 5042, Australia; 6Atherosclerosis Laboratory, ANZAC Research Institute, Concord Repatriation General Hospital, University of Sydney, Sydney, NSW 2000, Australia; 7Department of Cardiology, Concord Repatriation General Hospital, Hospital Road, Concord, Sydney, NSW 2139, Australia

**Keywords:** public health messaging, cardiology, acute coronary syndrome

## Abstract

Objective: The National Heart Foundation’s Warning Signs Campaign (2009–2013) aimed to raise awareness amongst the public of Acute Coronary Syndrome (ACS), encouraging people to recognise suggestive symptoms and seek immediate medical attention. This study explores the impact of the campaign on the characteristics of patients presenting to hospitals around Australia with ACS. Design: Retrospective cohort analysis Setting: 10 Australian Hospitals recruiting for the CONCORDANCE registry continuously throughout the campaign period. Participants: Patients presenting with ACS to hospitals before, during and after the campaign ran in their jurisdiction. Main Outcome Measures: Whether an ambulance was called, time between onset of symptoms to first medical contact, as well as time between onset of symptoms to primary percutaneous intervention or lysis. Results: Time to first medical contact did not improve during or post-campaign for NSTEACS medical hours (IQI) 1.6 (0.5–4.8) pre, 2.2 (0.7–7.6) during, 2 (0.7–6.9) post (*p* < 0.001) or STEMI, 1.1 (0.4–3.5) pre, 1.6 (0.6–5.1) during, 1.4 (0.5–4.3) post (*p* = 0.0113). In STEMI, time from symptom onset to pPCI (*p* = 0.256) and time to lysis (*p* = 0.387) were also unchanged. The proportion of ambulance arrivals remained stable (pre 55% vs. during 58%, *p* = 0.493). Patients presenting during the campaign were more likely to be born in Australia 56% pre, 69% during, 68% post (*p* < 0.001), to report English as a first language 67% pre, 84% during, 79% post (*p* < 0.001), and had lower likelihood of prior MI or revascularization but greater likelihood of cardiovascular risk factors compared to those presenting prior. Conclusion: Among patients with ACS, we detected no increase in proportion of ambulance presentations nor earlier presentations among NSTEACS or STEMI during the campaign. There was an increase in the proportion of patients for whom English was the first language and those without a prior cardiac history but with cardiovascular risk factors, suggesting that the campaign impacted preferentially on certain strata in the community.

## 1. Introduction

The National Heart Foundation’s ‘Heart Attack Warning Signs’ Campaign (The Campaign) was an Australian national public health campaign, which ran at varying intervals between 2009 and 2013, with the aim of raising awareness amongst the public of symptoms of Acute Coronary Syndrome (ACS) [1]. The campaign specifically aimed to equip the public with the confidence to recognise and act on these symptoms by providing a clear “heart attack action plan”, and highlighted the importance of seeking prompt medical attention by calling an ambulance [2,3]. Multiple media platforms such as flyers, radio and television advertisements were used to reach a maximum number of people. This study sought to evaluate the success of The Campaign by exploring the characteristics of patients presenting to hospital with acute coronary syndrome before, during and after the campaign period. We hypothesised that The Campaign would result in: (1) a higher proportion of those patients presenting via ambulance transportation, and (2) earlier presentations of patients with acute coronary syndrome.

## 2. Materials and Methods

The CONCORDANCE registry data, that collected data on patients presenting with documented ACS from 43 hospitals across the country from 2009 to 2018 [4]. To assess the impact of the campaign, 3 time periods were set—pre-campaign, during and post campaign. Patients of hospitals, where they recruited more than 5 patients at all 3 time points, were included in the analysis. Baseline characteristics and outcomes were compared between these time periods. For this study, patients with ACS were analysed, and subsample analysis was performed on those with STEMI.

### Statistical Analysis

Characteristics analysed included age, sex, first language and comorbidities. Outcomes included whether an ambulance was called, time between onset of symptoms and first medical contact, as well as time between onset of symptoms and primary percutaneous coronary intervention (PCI) or fibrinolytic therapy for patients with STEMI.

Univariable comparisons of patient characteristics at baseline and outcomes between patients presenting pre-campaign, during and post-campaign were performed.

Categorical variables were reported in numbers and percentages, and the Rao–Scott chi-square test was used to test the difference while adjusting for the clustering effect of the hospitals. Continuous variables were reported in means and standard deviations (SD) for normally distributed variables, and the univariable logistic regression model within the framework of the generalized estimating equation (GEE) was used to test the difference while adjusting for the clustering effect of the hospitals. Medians and interquartile intervals (IQI) were reported for skewed variables, and the Kruskal–Wallis test was used to test for the difference. For continuous and binomial variables, univariable trend analyses were performed. Linear regression within the framework of GEE was used for normally distributed continuous variables, the Mann–Kendall test was used for the skewed continuous variables, and the Cochran–Armitage test was used for the binomial variables. *p*-values less than 0.05 were considered significant. All analyses were conducted using SAS 9.4 (SAS Institute Inc., Cary, NC, USA).

## 3. Results

In total, 4339 patients with confirmed ACS from 10 hospitals were included in the analysis (Table 1). There were 805 admissions pre-campaign, 1265 during and 2269 in the post-campaign group.

### 3.1. Ambulance Presentations and Treatment Times (Table 2)

The proportion of patients arriving by ambulance did not change during the campaign (55% pre, 58% during, 58% post) (p_diff_ = 0.493).

Time to first medical contact did not improve during or post-campaign for NSTEACS median hours (IQI) 1.6 (0.5–4.8) pre, 2.2 (0.7–7.6) during, 2 (0.7–6.9) post (p_trend_ = 0.0155) or STEMI, 1.1 (0.4–3.5) pre, 1.6 (0.6–5.1) during, 1.4 (0.5–4.3) post (p_trend_ = 0.2312).

Among STEMI patients, the time from symptom onset to pPCI did not fall median hours (IQI) 3.7 (2.6–7.4) pre, 3.9 (2.7–10.3) during, and 3.5 (2.3–8.9) post (p_trend_ = 0.5924). Similarly, there was no improvement seen in time to lysis, 2.3 (1.4–3.9) pre, 2.6 (1.6–5.5) during, 2.6 (1.4–5.5) post (p_trend_ = 0.4865) (Table 2).

**Table 2 ijerph-19-10700-t002:** Ambulance presentations and times to treatment.

Variable	Statistics/Levels	Pre WS n (%) n = 805	During WS n (%) n = 1265	Post WS n (%) n = 2269	Total n (%) n = 4339	*p*-Value	*p*-Value for Difference	*p*-Value for Trend
Ambulance called		425 (55.1)	696 (58)	1249 (58)	2370 (57.5)	0.4926	0.4926	0.2045
Time to first medical contact for NSTEACS (h) (n = 4277)	Median (IQI)	1.6 (0.5, 4.8)	2.2 (0.7, 7.6)	2 (0.7, 6.9)	2 (0.7, 6.8)	<0.001	<0.001	0.0155
Time to first medical contact for STEMI (h) (n = 1205)	Median (IQI)	1.1 (0.4, 3.5)	1.6 (0.6, 5.1)	1.4 (0.5, 4.3)	1.4 (0.5, 4.4)	0.0113	0.0113	0.2312
STEMI Onset to PPCI time (h) n = 478	Median (IQI)	3.7 (2.6, 7.4)	3.9 (2.7, 10.3)	3.5 (2.3, 8.9)	3.7 (2.5, 9.3)	0.2566	0.2566	0.5924
STEMI onset to lysis time (h) n = 417	Median (IQI)	2.3 (1.4, 3.9)	2.6 (1.6, 5.5)	2.6 (1.4, 5.5)	2.5 (1.5, 5.3)	0.3866	0.3866	0.4865

### 3.2. Patient Characteristics during the Campaign

Patients who presented to hospital during the campaign were significantly more likely to be born in Australia (56% pre, 69% during, 68% post (p_diff_ < 0.001) and were more likely to report English as a first language 67% pre, 84% during, 79% post (p_diff_ < 0.001) compared to those presenting prior. They were less likely to have had prior MI (p_diff_ = 0.001) and previous PCI (p_diff_ < 0.001), but were more likely to have risk factors for coronary disease; specifically, a family history (p_diff_ < 0.001), hypertension (p_diff_ < 0.001) and dyslipidaemia (p_diff_ = 0.024) (Table 3).

The majority of all diagnoses were NSTEMI (48%), the proportion of ACS patients presenting with NSTEMI increased during and after the campaign (42% pre, 47% during, 51% post) (p_trend_ < 0.001) at the expense of STEMI (34% pre, 31% during, 24% post) (p_trend_ < 0.001), which may be partly explained by the role out of high sensitivity troponin testing during this period.

## 4. Discussion

In this detailed analysis of over 4300 patients presenting to Australian hospitals before, during and after a national initiative to improve awareness of symptoms of acute coronary events and actions to be taken in response to these symptoms, we found no change in the proportion of patients who called an ambulance, and no fall in time from symptom onset to hospital presentation or treatment.

Patients presenting to hospital during the campaign period were more likely to be born in Australia, and to report English as a first language when compared to those presenting before the Campaign. In contrast to previous studies, this effect did not appear to tail off with time, with the demographics of those presenting to hospital in the years following the campaign resembling those who had presented during and in the 6 months following. These findings suggest that, whilst the demographic of patients presenting may have been influenced by the campaign, some of the key messages did not affect the desired behaviours.

Studies looking at the effectiveness of the Warning Signs Campaign have found that patients reported both an increased awareness of “heart attack” symptoms, as well as increased confidence to act or call an ambulance as a result of the campaign [3,5]. It has been shown that language proficiency, amongst other socioeconomic factors, is a strong predictor of health literacy [6,7,8]. Members of the public without English as a first language may, therefore, have had less meaningful exposure to the campaign message. It has also been reported elsewhere that public health campaigns appear to have the least influence amongst those from a lower socioeconomic demographic, with lower levels of education, although our study did not directly examine these variables [9,10,11]. This suggests that the health education provided by such campaigns may not be reaching the most at-risk populations, with the greatest potential to benefit.

The duration of effect of public health campaigns appears to vary depending on several variables including the message, the audience and surrounding social norms mitigating behaviour [12]. Sly et al. found that defunding of the tobacco-use prevention program resulted in a return of pro-tobacco attitudes within 6 months [13]. A Japanese study looking at a public health campaign to encourage home deaths in palliative care found that there was only a small short-term improvement in public perception of home death, which was lost at 6-month follow-up [14]. A 2014 paper describing the effect of the “Time to Change” Anti-Stigma campaign, aimed at improving attitudes to mental health in the UK, saw improved attitudes 2 years post-campaign. However, they described multiple possible contributing influences including the corresponding release of a book on living with Bipolar Disorder by a public figure that received wide media attention [15].

The effect of the Warning Signs Campaign appeared to last longer, as measured by the demographic influenced, than previous public health campaigns. Rudolph et al. cited the use of multiple media strategies, such as was used in the Warning Signs campaign, as an effective method to amplify a campaign’s message [16]. It has also been noted that public health campaigns are less likely to achieve success when their message is faced by contradictory messaging, such as those encouraging smoking cessation in the face of tobacco advertisements and cultural norms [16]. The use of ambulance services is already well established in Australia, and so messaging around appropriate use is unlikely to face significant contest. With coronary artery disease being the leading cause of death in Australia, the topic of the campaign is likely to have resonated with many [17].

Certain factors have been associated with effective public health messaging. Studies examining health communication during the COVID-19 pandemic found that directives were better received coming from scientific or medical associations, over government sources [18,19]. Strategies that highlight personal responsibility for health, and empower the public to take specific actions, have also demonstrated increased public response [20,21]. Interestingly, Wise et al. highlighted risk perception as a predictor of protective health behaviours, as did a Nigerian study into compliance to COVID-19 public health directives [22,23]. The Warning Signs Campaign was produced by a credible medical association (The National Heart Foundation) and certainly provided specific instructions on how to respond to the outlined symptoms. However, in contrast, we found that it was the lower risk group that were more likely to respond.

One limitation of our study is that we were limited to 10 hospitals which collected data continuously before, during and after the campaign. Nonetheless, these sites were located across five jurisdictions, increasing generalizability, and there was no systematic bias in the identification and inclusion of these hospitals in this study. The numbers of patients enrolled during and after the Campaign were greater than pre-Campaign. We do not believe this detracts from our findings, as it just reflects the recruitment time period of the CONCORDANCE registry relative to the Campaign in these hospitals. Some people may take some self-provided drugs before they call the ambulance or are admitted to the hospital, and this time from symptom to self-drug may be a good measure of the response to Warning Signs Campaign; unfortunately, these data were not available in this study.

## 5. Conclusions

In conclusion, our analysis suggests that the Warning Signs Campaign appears to have been an ineffective intervention, in that it did not impact on its primary objectives of increasing the proportion of patients calling an ambulance, or shortening the time to hospital treatment among patients with an ACS.

The Campaign did have some influence on public health-seeking behaviour, with the largest effect amongst English-speaking patients. Arguably, this demographic is more likely to have had pre-existing better health literacy, and more able to understand and to respond to the campaign’s message. This information should be used to inform strategies of future campaigns to augment their effectiveness and better target the most at-risk populations.

## Figures and Tables

**Table 1 ijerph-19-10700-t001:** Hospital characteristics.

Variable	Statistics/Levels	Total n (%) n = 10 Hospitals
No of beds in each hospital		
	Quartile 1	367
	Median	500.5
	Quartile 3	665
Cathlab available		7 (70)
Hospital in urban area		6 (60)
State	NSW	5 (50)
	NT	1 (10)
	QLD	2 (20)
	SA	1 (10)
	VIC	1 (10)

**Table 3 ijerph-19-10700-t003:** ACS-Baseline characteristics of ACS patients presenting pre, during and post-campaign.

Variable	Statistics/Levels	Pre WS n (%) n = 805	During WS n (%) n = 1265	Post WS n (%) n = 2269	Total n (%) n = 4339	*p*-Value for Difference	*p*-Value for Trend
Age	Mean (SD)	65.2 (14.1)	64.6 (13.5)	65.8 (13.7)	65.3 (13.7)	0.1245	0.7093
Sex	M	556 (69.1)	908 (71.8)	1565 (69)	3029 (69.8)	0.1521	0.5772
Country of birth	Australia	447 (55.5)	866 (68.5)	1546 (68.1)	2859 (65.9)	<0.0001	
	New Zealand	12 (1.5)	32 (2.5)	54 (2.4)	98 (2.3)		
	Great Britain	50 (6.2)	106 (8.4)	146 (6.4)	302 (7)		
	Other European	128 (15.9)	97 (7.7)	210 (9.3)	435 (10)		
	South Asian	29 (3.6)	19 (1.5)	52 (2.3)	100 (2.3)		
	Chinese	11 (1.4)	5 (0.4)	24 (1.1)	40 (0.9)		
	Other Asian	34 (4.2)	34 (2.7)	69 (3)	137 (3.2)		
Language	English is the first language	541 (67.2)	1057 (83.6)	1798 (79.2)	3396 (78.3)	<0.0001	<0.0001
Grace Risk Score (Fox)	Quartile 1	83.1	82.2	83.3	83.2	0.0926	0.1686
	Median	106.6	104	104.8	105		
	Quartile 3	133.2	127.5	128.6	129.1		
Prior MI		273 (33.9)	374 (29.6)	767 (33.8)	1414 (32.6)	0.0011	0.4775
Prior heart failure		74 (9.2)	98 (7.7)	227 (10)	399 (9.2)	0.3611	0.2106
Previous angiogram positive for coronary artery disease		315 (39.1)	412 (32.6)	937 (41.3)	1664 (38.3)	<0.0001	0.0142
Previous percutaneous coronary intervention		161 (20)	212 (16.8)	533 (23.5)	906 (20.9)	<0.0001	0.0013
Previous coronary artery bypass graft		117 (14.5)	166 (13.1)	307 (13.5)	590 (13.6)	0.1306	0.6024
Chronic renal failure		74 (9.2)	116 (9.2)	256 (11.3)	446 (10.3)	0.1142	0.0410
Previous stroke/transient ischemic attack		62 (7.7)	102 (8.1)	198 (8.7)	362 (8.3)	0.7365	0.3238
Diabetes		229 (28.4)	340 (26.9)	667 (29.4)	1236 (28.5)	0.0742	0.3604
Hypertension		469 (58.4)	768 (60.9)	1544 (68.3)	2781 (64.3)	<0.0001	<0.0001
Dyslipidaemia		436 (54.6)	734 (58)	1409 (62.3)	2579 (59.6)	0.0243	<0.0001
Family history of coronary heart disease		207 (25.7)	478 (37.8)	705 (31.1)	1390 (32)	<0.0001	0.2210
Diagnosis	STEMI	274 (34)	395 (31.2)	546 (24.1)	1215 (28)	<0.0001	<0.0001
	NSTEMI	339 (42.1)	589 (46.6)	1145 (50.5)	2073 (47.8)		<0.0001
	UA	192 (23.9)	281 (22.2)	578 (25.5)	1051 (24.2)		0.1492

Abbreviations: MI = myocardial infarction. NSTEACS = Non-ST-elevation acute coronary syndrome (unstable angina, NSTEMI). NSTEMI = Non-STE-elevation myocardial infarction. STEMI = ST-elevation myocardial infarction. H = hours. Cathlab = Catheter laboratory.

## Data Availability

Not applicable.

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
