# Peer review of "Impact of the “Warning Signs Campaign” on Characteristics of Patients Presenting with Acute Coronary Syndrome (ACS) to Hospitals"

_ijerph, 2022, doi:10.3390/ijerph191710700_

Round 1

Reviewer 1 Report

Redwood et al conducted a retrospective cohort analysis into the impact of the campaign on the characteristics of patients presenting with ACS to hospitals around Australia, and they found that Australians accounted for large proportions of patients presenting during the campaign, who mainly spoke English as a first language, yet no improvements in time to first medical contact were found during or post- campaign for NSTEMI, and in time from symptom onset to pPCI and time to lysis for STEMI. This study was retrospectively and simply analyzed, and of limited reference significance for other countries. Its data were not comprehensively and reasonably displayed, more importantly, and conclusions “Our study suggests the Warning Signs Campaign did influence public health- seeking behaviour, with the largest effect amongst lower risk, English speaking patients” were over-interpreted.

ACS refers to any group of clinical symptoms compatible with acute myocardial ischemia and covers the spectrum of clinical conditions ranging from unstable angina (UA) to NSTEMI to STEMI, in the Abstract section, the authors seemed to neglect those with UA. Moreover, the Abstract should be re-written, regarding Design, Setting and Participants.

In the last paragraph of Introduction Section, it was hypothesized that Campaign would result in earlier presentations of patients with acute coronary syndrome, and a higher proportion of those patients presenting via ambulance transportation, however, both hypotheses were not supported by data displayed in Table 3, as evidenced unchanged percentage of ambulance call, even prolonged time to first medical contact or time from symptom onset among NSTEMI or STEMI patients, which might contradict with the conclusion “Warning Signs Campaign did influence public health-seeking behaviour”.

The authors stated that “Patients presenting to hospital during the warning signs campaign period reflected a younger, lower- risk group, who were more likely to be born in Australia”, however, the age and Grace risk score discrepancies between difference stages were rather small, take Grace risk score for example, no significant differences were found between groups, thus data presented can not support the authors’ conclusions.

Some data in Table 2 may be better presented as Median (Quartile 1, Quartile3). Whether there were any differences between patients’ baseline characteristics in different hospitals or states? For a single hospital or state, whether patients’ baseline characteristics changed in different period?

Some people may take some self-provided drugs before they call the ambulance or are admitted to the hospital, and this time from symptom to self-drug is a very good response to Warning Signs Campaign, unfortunately, the data was missing in this study.

Title Impact of the Warning Signs Campaign on Characteristics of Patients Presenting with Acute Coronary Syndrome to Hospitalsmight be more appropriate

Please kindly provide the Registry Number of CONCORDANCE Registry

Author Response

Redwood et al conducted a retrospective cohort analysis into the impact of the campaign on the characteristics of patients presenting with ACS to hospitals around Australia, and they found that Australians accounted for large proportions of patients presenting during the campaign, who mainly spoke English as a first language, yet no improvements in time to first medical contact were found during or post- campaign for NSTEMI, and in time from symptom onset to pPCI and time to lysis for STEMI. This study was retrospectively and simply analysed, and of limited reference significance for other countries. Its data were not comprehensively and reasonably displayed, more importantly, and conclusions “Our study suggests the Warning Signs Campaign did influence public health- seeking behaviour, with the largest effect amongst lower risk, English speaking patients” were over- interpreted.

Response:  We have made some alterations to the presentation of our data, showing the important outcomes of ambulance presentations and times to treatment first and then patient demographics after this.  We believe this improves the display of our results.  In addition we have modified our conclusions to limit over interpretation in both the abstract and the discussion. 

ACS refers to any group of clinical symptoms compatible with acute myocardial ischemia and covers the spectrum of clinical conditions ranging from unstable angina (UA) to NSTEMI to STEMI, in the Abstract section, the authors seemed to neglect those with UA. Moreover, the Abstract should be re-written, regarding Design, Setting and Participants.

Response:  The data cited in the abstract as reflecting NSTEMI actually were derived from NSTEACS which include UA.  This has been corrected.  As requested the structure of the abstract has been revised.

In the last paragraph of Introduction Section, it was hypothesised that Campaign would result in earlier presentations of patients with acute coronary syndrome, and a higher proportion of those patients presenting via ambulance transportation, however, both hypotheses were not supported by data displayed in Table 3, as evidenced unchanged percentage of ambulance call, even prolonged time to first medical contact or time from symptom onset among NSTEMI or STEMI patients, which might contradict with the conclusion “Warning Signs Campaign did influence public health-seeking behaviour”.

Response:  We respectfully suggest that while the our data did not show an improvement in ambulance presentations or times to treatment, there was some impact on public health seeking behaviour reflected in an increasing proportion of presentations among patients of a particular demographic.  This message has been clarified in the conclusions.  

The authors stated that “Patients presenting to hospital during the warning signs campaign period reflected a younger, lower- risk group, who were more likely to be born in Australia”, however, the age and Grace risk score discrepancies between difference stages were rather small, take Grace risk score for example, no significant differences were found between groups, thus data presented can not support the authors’ conclusions.

Response:  We agree with these comments and have altered our description of these patients to more accurately match their demographics. 

Some data in Table 2 may be better presented as Median (Quartile 1, Quartile3). Whether there were any differences between patients’ baseline characteristics in different hospitals or states? For a single hospital or state, whether patients’ baseline characteristics changed in different period?

Response: With the limited number of hospitals available for this analysis we did not have the ability to look at differences between states. 

Some people may take some self-provided drugs before they call the ambulance or are admitted to the hospital, and this time from symptom to self-drug is a very good response to Warning Signs Campaign, unfortunately, the data was missing in this study.

Response:  This pertinent observation has been included as a limitation in the discussion.

Title “Impact of the Warning Signs Campaign on Characteristics of Patients Presenting with Acute Coronary Syndrome to Hospitals” might be more appropriate

Please kindly provide the Registry Number of CONCORDANCE registry

Response: The title has been changed as requested.  This study was not registered as it was initiated before registration of registries was common or accepted practice. 

Reviewer 2 Report

In this article authors discuss about the role of warning signs campaign in Australian population. I have the following concerns

1. The campaign was from 2009-2013? What is the reason for delay in completing the research and manuscript. Current standards of ACS care is much different

2. In all the tables authors have presented p value for 3 separate variable.  Please elaborate on what was compared with what ? were variables analyzed between the groups 

3. Please expand the abbreviations below the table

Author Response

In this article authors discuss about the role of warning signs campaign in Australian population. I have the following concerns

  1. The campaign was from 2009-2013? What is the reason for delay in completing the research and manuscript. Current standards of ACS care is much different.

Response: we agree it has taken some time to produce this analysis, due to competing priorities and the impact of the COVID pandemic on time for academic work over the last two years.  However we believe the relevant practices with regards to early presentation and treatment of acute coronary syndromes have not changed appreciably.

  1. In all the tables authors have presented p value for 3 separate variable.  Please elaborate on what was compared with what ? were variables analyzed between the groups.

Response: when comparing 3 groups the P value test for trend is used.  A significant value means that at least one of the groups differs from the other two.  

  1. Please expand the abbreviations below the table

Response: this has been performed as requested

Reviewer 3 Report

Dear authors,

Your study is fairly constructed and conducted. The results are clearly presented and the conclusions are clear. However, the potential interest in reading your article might not be the expected one, due to the limitations to Australian population. Also, the reference section is not so dense, and there is a lack of new references, from 2020-2022, maybe you should add a section in the discussions in which you explain that you have searched Web of Science using keywords and you have found that similar studies are few.

Good luck in improving the Discussions and finding other studies to discuss/compare your results. 

Best regards, 

Author Response

Dear authors,

Your study is fairly constructed and conducted. The results are clearly presented and the conclusions are clear. However, the potential interest in reading your article might not be the expected one, due to the limitations to Australian population. Also, the reference section is not so dense, and there is a lack of new references, from 2020-2022, maybe you should add a section in the discussions in which you explain that you have searched Web of Science using keywords and you have found that similar studies are few.

Response:  the references have been updated as requested.